# Few-Shot Image Classification of Crop Diseases Based on Vision–Language Models

**DOI:** 10.3390/s24186109

**Published:** 2024-09-21

**Authors:** Yueyue Zhou, Hongping Yan, Kun Ding, Tingting Cai, Yan Zhang

**Affiliations:** 1School of Information Engineering, China University of Geosciences, Beijing 100083, China; doublemoon171@gmail.com (Y.Z.); caitingting0903@163.com (T.C.); cangmiszy@gmail.com (Y.Z.); 2State Key Laboratory of Multimodal Artificial Intelligence Systems (MAIS), Institute of Automation, Chinese Academy of Sciences, Beijing 100190, China; kun.ding@ia.ac.cn

**Keywords:** few-shot learning, crop disease classification, vision–language models, attention mechanisms

## Abstract

Accurate crop disease classification is crucial for ensuring food security and enhancing agricultural productivity. However, the existing crop disease classification algorithms primarily focus on a single image modality and typically require a large number of samples. Our research counters these issues by using pre-trained Vision–Language Models (VLMs), which enhance the multimodal synergy for better crop disease classification than the traditional unimodal approaches. Firstly, we apply the multimodal model Qwen-VL to generate meticulous textual descriptions for representative disease images selected through clustering from the training set, which will serve as prompt text for generating classifier weights. Compared to solely using the language model for prompt text generation, this approach better captures and conveys fine-grained and image-specific information, thereby enhancing the prompt quality. Secondly, we integrate cross-attention and SE (Squeeze-and-Excitation) Attention into the training-free mode VLCD(Vision-Language model for Crop Disease classification) and the training-required mode VLCD-T (VLCD-Training), respectively, for prompt text processing, enhancing the classifier weights by emphasizing the key text features. The experimental outcomes conclusively prove our method’s heightened classification effectiveness in few-shot crop disease scenarios, tackling the data limitations and intricate disease recognition issues. It offers a pragmatic tool for agricultural pathology and reinforces the smart farming surveillance infrastructure.

## 1. Introduction

In the era of rapid advancements in agricultural technology, the capability to swiftly and accurately classify and identify crop diseases emerges as a cornerstone in safeguarding food security and propelling productivity enhancements. The traditional manual diagnosis methods, being time-consuming and prone to subjective biases, have given way to computer-assisted automated classification techniques. The early automated classification methods heavily relied on digital image processing and conventional machine learning algorithms [1], which are highly dependent on expert knowledge and extensive sample validation, thereby limiting their efficacy in complex field environments.

The advent of deep learning has prompted researchers to adopt deep neural network models that directly input images for disease classification [2]. This innovation not only simplifies background noise handling but also eliminates the need for manual layer-by-layer design and feature extraction, significantly economizing the resources and enhancing the accuracy in complex disease recognition scenarios. Deep learning models typically necessitate large quantities of labeled training data. However, in agriculture, acquiring and annotating crop disease images is arduous and costly. This scarcity in training samples hampers the widespread adoption of these methods. Moreover, the prevailing research predominantly focuses on unimodal image analysis, overlooking the vast potential of multimodal information in enhancing model generalizability and accuracy, especially with limited samples available. This oversight becomes particularly striking in intricate agricultural disease identification. Given the subtle differences between similar disease manifestations and the need for robust performance under different real-world conditions, new analytical strategies are required to solve this issue. Our work aims to bridge this gap by harnessing the potential of integrating image and text modalities, thus overcoming the limitations of the current unimodal approaches in accurately identifying complex diseases with limited data.

Recently, pre-trained VLMs have made significant strides in Computer Vision, offering a groundbreaking approach to tackle the few-shot classification challenges in this field [3]. By integrating both image and text data, VLMs excel at understanding and representing complex semantic content by jointly modeling images and their corresponding textual descriptions, thereby enhancing the generalizability in few-shot learning. Compared to the conventional unimodal image analysis techniques, VLMs can simultaneously consider both image and text data, enhancing the semantic correlation between them and thereby improving the richness and distinctiveness of the features. In natural image domains, especially in few-shot scenarios, pre-trained VLMs on large-scale datasets have demonstrated remarkable cross-domain transferability [4]. Nevertheless, their application in specialized image contexts, like crop disease, remains unknown.

This study is dedicated to innovatively enhancing the few-shot crop disease classification accuracy by incorporating pre-trained VLMs. Broadly speaking, we establish a framework named VLCD (**V**ision–**L**anguage model for **C**rop **D**isease classification) for few-shot crop leaf disease classification by introducing Qwen-VL [5] and CLIP (Contrastive Language–Image Pre-training) [3], combining image information and attention mechanisms. Specifically, considering that different crop leaf diseases have different color texture visual characteristics, we first select representative samples from each category based on the image color and texture features. Next, we employ Qwen-VL to generate detailed text descriptions for these representative images, serving as prompt texts, which will be input to CLIP’s text encoder to extract the text features. Leveraging image information in prompt text generation enables the effective capture of subtle disease characteristics and crucial details. To further optimize the text features, we utilize cross-attention in the training-free VLCD to integrate multiple prompt text vectors for each category (i.e., intra-class prompt text vectors) and employ SE Attention in the training-required VLCD-T (**VLCD**-**T**raining) to process the prompt text vectors between different categories (i.e., inter-class prompt text vectors) to highlight the key text features. These text vectors ultimately serve as the final classifier weights. Additionally, to shorten the training period, we employ the key-value cache model [6] and the Prior Refinement module [7] to fine-tune CLIP, adapting to the agricultural domain. The empirical results demonstrate the excellent performance of our method in few-shot crop leaf disease classification. Our method offers a fresh perspective to alleviate data scarcity and improve the agricultural disease classification in few-shot scenarios. With reference to the ongoing debate regarding model-centric AI and data-centric AI [8], our work belongs to the category of model-centric AI, highlighting the significance of further developing model-centric approaches [9].

The main contributions of this study are as follows:Enhancing few-shot crop leaf disease classification accuracy by multimodal integration: By fine-tuning CLIP, we integrate image and text information, providing a paradigm for the development of multimodality in the agricultural field. The experimental results show that our method exhibits excellent classification performance in few-shot scenarios, effectively solving the challenge of data scarcity.Fine-grained disease feature description driven by VLM: We innovatively leverage Qwen-VL’s fine-grained recognition to generate detailed textual descriptions of crop diseases as prompt texts from a set of infected crop leaf images to assist in generating discriminative classifier weights. This enhances the model’s sensitivity and accuracy in identifying complex diseases from the images of infected crop leaves.Enhancing key textual features by cross-attention and SE Attention: In the process of processing prompt texts, we use cross-attention and SE Attention, respectively, in the training-free and training-required modes to guide the model’s attention to important textual features. By dynamically adjusting the weights of crucial prompt texts’ features, we effectively improve the quality of the model’s classification weights.

## 2. Related Work

### 2.1. Development of Crop Disease Image Classification Technology

Since the 1980s, researchers have predominantly employed traditional machine learning techniques like Support Vector Machines (SVMs) and K-Nearest Neighbor (KNN) classifiers for crop disease image classification [1]. However, these methods are dependent on high-quality, large-scale training data and have limitations in feature design. The significant enhancement in accuracy achieved by Deep Neural Networks (DNNs) within the realm of image recognition has prompted agricultural researchers to adopt DNNs for classification tasks. Illustratively, Irmak and Saygili [10] proposed an automatic classification method for tomato leaf diseases based on a deep Convolutional Neural Network (CNN), which combined traditional machine learning and deep learning to significantly improve the classification accuracy. Ferentinos [11] used a CNN to identify 58 diseases of 25 plant species with an accuracy of up to 99.53%.

Crop disease classification is a fine-grained problem due to factors like uneven lighting, occlusions, complex backgrounds, and disease progression. In this regard, Guo et al. [12] employed a Bayesian algorithm to integrate the texture and color features for the effective identification of four distinct crop diseases, thereby enhancing the classification accuracy. Zhang et al. [13] developed an intelligent recognition application for cucumber leaf disease, employing k-means clustering to classify the disease spots and extract features, achieving an accuracy rate of 85.7%.

However, due to the broad crop diversity and the large differences between varieties, it is difficult to collect large-scale standardized samples. To address the data scarcity, transfer learning has emerged as an effective auxiliary tool for neural network training. By leveraging pre-learned model parameters, transfer learning accelerates the learning process of new models without necessitating training from scratch. Kaya et al. [14] investigated and substantiated the utility of transfer learning models in facilitating crop classification. Bai et al. [15] introduced a rice disease detection method grounded in multi-source data and transfer learning, significantly boosting the disease recognition accuracy. Transfer learning eases the demand for extensive training samples to some extent, yet it does not resolve the overfitting issue that arises when models are trained with limited data. This has prompted researchers to explore few-shot learning, aiming for deep networks to learn effectively from a small number of samples.

In recent years, few-shot learning has found applications in crop disease classification and recognition. Li et al. [16] proposed a semi-supervised few-shot learning approach to tackle plant leaf disease identification, demonstrating its superiority through experiments conducted with limited annotated training data. Nuthalapati and Tunga [17] presented a method for automatic agricultural pest and disease image classification using few-shot learning, and also validated the effectiveness across multiple datasets, providing robust support for augmenting agricultural yields. Despite the demonstrated applicability and some achievements in crop disease classification, few-shot learning is inherently constrained by its reliance on limited training data, often struggling to generalize effectively to unseen data, resulting in unstable and inaccurate predictions on unknown instances.

From traditional machine learning algorithms to deep learning models, most methods mainly focus on a single image modality for analysis. Different from the previous methods, we innovatively introduced VLMs to solve the existing problems. VLMs adeptly fuse image and text information, yielding semantically enriched feature representations, ultimately enhancing the model learning efficiency and generalization capabilities under few-shot scenarios. By introducing pre-trained VLMs, we aim to enhance the accuracy of few-shot crop leaf disease classification.

### 2.2. Vision–Language Models and Fine-Tuning

Over the past two years, VLMs have gained prominence, finding widespread application in zero-shot and few-shot tasks [18]. Tailored for image–text data, VLMs encompass both image and text encoders along with fusion mechanisms, fostering associations between the two modalities through joint learning. Following extensive pre-training, VLMs can greatly improve various visual tasks. During the development of VLMs, CLIP [3] represents a significant stride forward in the integration of vision and language. CLIP employs an image–text contrastive objective, gauging the image–text similarity via dot products between their respective embeddings. Harnessing contrastive learning, CLIP achieves the alignment of the image–text features and zero-shot prediction, excelling across various visual tasks [19,20]. To adapt CLIP to specific tasks and facilitate domain transfer, researchers have devised numerous fine-tuning strategies. For example, CoOp [21] enhances the few-shot classification performance by optimizing continuous prompt contexts. CoCoOp [22] introduces dynamic conditional prompting to bolster the generalization capabilities. KgCoOp [23], innovated by Yao et al., aims to minimize the prompt variance while preserving the textual knowledge to tackle unseen categories. These methods achieve CLIP fine-tuning through prompt tuning. Additionally, adapter-based tuning has also been explored [24]. TaskRes [25] introduces a task-independent adapter to decouple the prior knowledge of pre-trained models from task-specific knowledge. GraphAdapter [26] leverages a bimodal knowledge graph to enhance the few-shot learning capabilities in VLMs by considering the category correlations between the visual and textual modalities, resulting in a more effective classifier.

However, these methods often necessitate additional training, thereby increasing the time and computational resource demands. In response, Lu et al. [27] explored the potential of ensemble learning with pre-trained VLMs, proposing customized ensemble strategies for different scenarios that can improve the model performance without requiring additional training. Tip-Adapter [6] introduces a key-value cache model that enhances CLIP’s accuracy in few-shot image classification without requiring any extra training. On this basis, APE [7] introduces an adaptive feature refinement mechanism, which not only identifies and strengthens the visual feature channels that are more discriminative for the task but also promotes a closer alignment between the visual and textual representations, significantly improving the performance of CLIP on specific tasks. APE stands out for its precise core feature identification, reduced caching needs, sustained high classification performance, and boosted adaptability and generalization in few-shot scenarios thanks to its innovative cross-modal trilateral relationship analysis. Drawing from these successful experiences, our study aims to boost the few-shot crop leaf disease classification accuracy by fine-tuning CLIP.

### 2.3. Automatic Generation of Prompt Text

In downstream classification tasks, the input prompt texts play a pivotal role in the pre-trained VLM [21]. Lewis et al. [28] utilized the large pre-trained language model GPT (Generative Pre-trained Transformer) [29] to automatically generate diverse fine-grained text descriptions for each category by constructing domain-specific prompts. These automatically generated texts by GPT are then fed as prompts that are input to CLIP, significantly enhancing the classification performance. Creating domain-specific prompts usually demands manual intervention, requiring considerable effort in conceptualizing, designing, and refining templates. Conversely, the novel VLM, Qwen-VL [5], demonstrates superior multimodal capabilities, excelling in image captioning, question answering, image localization, and text reading. It is endowed with powerful visual understanding and fine-grained recognition capabilities through carefully designed visual receptors, input–output interfaces, a three-stage training process, and a multilingual multimodal cleaned corpus. We only need to input a disease image and a concise question and Qwen-VL’s fine-grained recognition capabilities can automatically capture the key details in leaf disease images and generate high-quality prompt texts without elaborate templates. This provides a significant advantage over those methods relying solely on GPT or other single-language models. Its unique strengths stem from deep learning on high-resolution images and meticulously annotated data, allowing Qwen-VL to precisely parse intricate visual features and generate detailed text descriptions. This not only improves the prompt text quality but also eliminates the laborious task of manual template design, drastically reducing costs. Additionally, because Qwen-VL is free and open-source, it presents a cost-effective alternative compared to models like GPT-4 [30], which require significant computational resources and associated costs. Moreover, Qwen-VL does not require fine-tuning for our specific use case; it can directly generate detailed text descriptions from images and simple questions using the pre-trained model, streamlining the process and reducing the need for additional training. To construct discriminative classifier weights, we harness Qwen-VL’s fine-grained recognition to generate high-quality prompt texts to support classification decisions. To balance efficiency and precision, we adopt a clustering strategy, generating descriptions only for the representative images within each category. This approach ensures comprehensive information while avoiding resource wastage from irrelevant images, achieving a balanced mix of efficiency and precision.

### 2.4. Attention Mechanism

An attention mechanism [31] emulates human attention, enabling models to focus on crucial information. It has been widely adopted in both Natural Language Processing (NLP) and Computer Vision (CV). In NLP, starting from machine translation, its applications have expanded to tasks like text generation, summarization, and question answering [32,33]. In CV, the attention mechanism facilitates tasks like image segmentation [34], video classification [35], visual question answering [36], and object tracking [37], directing the model’s focus to salient regions in images to enhance understanding. With the convergence of vision and language, attention mechanisms have been integrated into cross-modal models. Molloy et al. [38] employed self-attention to refine the key information from visual description texts, bolstering the generalization capacity of CLIP. In our study, we explore how attention mechanisms can be leveraged to accentuate the critical features within prompt texts. We innovatively combine cross-attention and lightweight SE (Squeeze-and-Excitation) Attention to optimize the feature weight distribution when the model processes the prompt text. Cross-attention [31] deals with multi-source or multimodal data, dynamically attending to and interacting with the relevant portions. In our study, it is applied to inter-class prompt text vectors, merging their textual features. SE Attention, originating from SENet [39], “squeezes” the global information of feature channels and generates weight vectors through “excitation” to adjust the channel weights, emphasizing the key features specific to the task. The SE Attention module serves as a plug-and-play mechanism that can be seamlessly integrated into models without significantly increasing their model complexity. Therefore, we utilize it to process inter-class prompt text vectors during training, highlighting the subtle discriminative features among the different disease categories. Cross-attention and SE Attention can be easily integrated into the model without significantly increasing the development costs or training time. By integrating these two mechanisms, we aim to optimize the allocation of feature weights when the model processes the prompt texts, thereby adjusting the model’s classification weights to enhance the accuracy.

## 3. Method

In this study, we improve the performance of few-shot crop disease classification by fine-tuning CLIP. The models we designed, VLCD and VLCD-T, are built upon APE (Adaptive Prior rEfinement) [7] and APE-T (APE-Training), respectively (Figure 1 (1)), incorporating two specialized designs tailored to the characteristics of crop disease classification. First, we devise an image-driven prompt text generation module to obtain detailed prompt texts on representative images of different categories. Specifically, we cluster images based on color and texture features, selecting representative image sets for each category from the training set. We then leverage Qwen-VL’s fine-grained recognition function to pose queries about these representative images, generating detailed textual descriptions. Ultimately, we summarize and organize the text descriptions of all categories, using them as prompt texts for the subsequent classification tasks (Figure 1 (2)).

Second, in the processing of prompt texts’ features, we introduce attention mechanisms to guide attention to important text feature vectors. Specifically, in the training-free VLCD mode, we use CLIP’s text encoder to process the prompt texts generated in Figure 1 (2), obtaining prompt text vectors. These vectors are then subjected to a cross-attention mechanism to derive model classification weights for each category (Figure 1 (3)). Subsequently, SE Attention is used to process inter-class prompt text vectors in the training-required mode VLCD-T, emphasizing subtle discriminative features between different disease categories (Figure 1 (4)). Cross-attention and SE Attention serve to enhance prompt texts’ features in the training-free and training-required modes, respectively.

To present the proposed method clearly, this section will follow the following sequence: Section 3.1 provides a detailed overview of the background work on which this study is based; Section 3.2 introduces the image-driven prompt text generation strategy; Section 3.3 describes the prompt text features based on cross-attention fusion in training-free mode; Section 3.4 describes the prompt text features’ enhancement based on SE Attention in training-required mode.

### 3.1. Background

#### 3.1.1. Zero-Shot CLIP

In CLIP zero-shot classification, a descriptive language prompt is created for each test class in a dataset of *C* categories, formatted as “A photo of a [CLASS]”, where [CLASS] is the specific class name. This prompt is encoded by CLIP’s text encoder into a prompt vector $W\in R^{C\times D}$, serving as the classifier weight matrix, where *D* is embedding dimension. The image encoder then extracts a *D*-dimensional visual embedding $f\in R^{D}$. Finally, CLIP uses their similarity to calculate the zero-shot classification logits as follows: (1)$$R_{fW}=fW^{⊺}\in R^{1\times C}.$$

Zero-shot classification is achieved by finding the class corresponding to the maximum value of $R_{fW}$. However, its accuracy is limited as it does not leverage data from new tasks. For this reason, works like Tip-Adapter [6] utilize a cache model to improve CLIP classification accuracy without extra training.

#### 3.1.2. Cache Model

Tip-Adapter [6] enhances CLIP’s few-shot classification by using a key-value cache model for quick adaptation with small sample sizes. Unlike zero-shot CLIP, it adds logits based on the similarity between test and training features.

Specifically, for a *C*-category dataset with *K* samples per class, the cache model consists of keys and values. Keys are visual features extracted by CLIP’s image encoder from the training images, expressed as $F\in R^{CK\times D}$. Values are one-shot vectors representing the category labels of these training images, expressed as $L\in R^{CK\times C}$.

For a test image, $f\in R^{D}$, its similarity with the training-set features, $F\in R^{CK\times D}$, is calculated as
(2)$$R_{fF}=\exp\left(-\beta \left(1-fF^{⊺}\right)\right),$$
where β is a smoothing scalar. $R_{fF}$ is regarded as weights to integrate L, and the query results are blended with the zero-shot prediction, $R_{fW}$, to form the final logits: (3)$$\mathrm{logits}=R_{fW}+\alpha R_{fF}L,$$
where α is a balancing factor. By using $R_{fF}$, the cache model can utilize the existing knowledge in the training data to classify the test images.

Furthermore, Tip-Adapter-F can further improve accuracy by fine-tuning the cache keys, but this demands extensive cache and numerous learnable parameters. Therefore, Zhu et al. [7] proposed training-free APE and training-required APE-T, which combine CLIP’s prior knowledge and introduce lightweight parameters to reduce computational resource. Our models build on APE and APE-T, and we provide a detailed overview below.

#### 3.1.3. APE

APE introduces a Prior Refinement (PR) module to analyze class differences and refine core knowledge, performing a feature selection operation in the channel dimension to identify a subset of features that accurately represent the data. The PR module extracts the most informative *E* channels for the three features, denoted as $f^{\prime }\in R^{E}$, $W^{\prime }\in R^{C\times E}$, and $F^{\prime }\in R^{CK\times E}$.

APE explores the relationships among test image features, prompt texts, and cache model features, as shown on the left side (a) of Figure 1 (1):The relationship between f and W, determined using Equation (Equation 1), represents the cosine similarity between the test image and the prompt texts.The relationship between $f^{\prime }$ and $F^{\prime }$ can be calculated using a similar method as described in Equation (Equation 2):
(4)$$R_{f^{\prime }F^{\prime }}=\exp\left(-\beta \left(1-f^{\prime }F^{\prime ⊺}\right)\right)\in R^{1\times CK},$$The relationship between $F^{\prime }$ and $W^{\prime }$ involves APE’s zero-shot CLIP prediction on training data, denoted as $F^{\prime }W^{\prime ⊺}$. To evaluate CLIP’s downstream recognition capability, the KL-divergence, $D_{KL}\left(|\right)$, between $F^{\prime }W^{\prime ⊺}$ and $L\in R^{CK\times C}$ is calculated as follows:
(5)$$R_{F^{\prime }W^{\prime }}=\exp\left(\gamma D_{KL}\left(F^{\prime }W^{\prime ⊺}|L\right)\right)\in R^{1\times CK},$$
where γ is a smoothing factor. $R_{F^{\prime }W^{\prime }}$ represents the score contribution of each training feature to the final prediction.

Finally, APE integrates all trilateral relations, and the final classification logits are computed as
(6)$$\mathrm{logits}=R_{fW}+\alpha R_{f^{\prime }F^{\prime }}\left(diag\left(R_{F^{\prime }W^{\prime }}\right)L\right),$$
where α is a balance factor and diag() denotes diagonalization. The second term improves the feature channel-based few-shot prediction with the reweighted cache model. Through the PR module and trilateral relationship analysis, APE effectively boosts CLIP’s few-shot classification performance.

#### 3.1.4. APE-T

In the training setting, as shown on the right side (b) of Figure 1 (1), APE-T freezes the cache model and trains only lightweight category residuals, $\mathrm{Res}\in R^{C\times E}$, along with the cache scores $R_{F^{\prime }W^{\prime }}$. These residuals are *C*-learnable embeddings, each corresponding to a category, optimizing the *E* feature channels during training. To maintain vision–language correspondence in the embedding space, APE-T applies Res to both textual features W and training-set features $F^{\prime }$. APE-T also explores trilateral relations:For $R_{fW}$ in Equation (Equation 1), APE-T first pads the *E*-channel Res into *D*-channels as W by filling the extra channels with zero. The padded Res, denoted as Pad(Res), is added to W, updating CLIP’s zero-shot prediction by the optimized textual features, formulated as
(7)$$R_{fW}=f*{\left(W+Pad\left(\mathrm{Res}\right)\right)}^{⊺}.$$For $R_{f^{\prime }F^{\prime }}$ in Equation (Equation 4), APE-T first broadcasts the *C*-embedding Res into CK as $F^{\prime }$ by repeating it for each category. Next, APE-T adds the expanded Res to $F^{\prime }$ element-wise, improving the cache model’s few-shot prediction by optimizing training-set features. This process is formulated as
(8)$$R_{f^{\prime }F^{\prime }}=\exp\left(-\beta \left(1-f^{\prime }{\left(F^{\prime }+Expand\left(\mathrm{Res}\right)\right)}^{⊺}\right)\right).$$For $R_{F^{\prime }W^{\prime }}$ in Equation (Equation 5), APE-T makes it learnable during training, enabling adaptive learning of optimal cache scores for different training-set features, determining their contribution to predictions.

Finally, the classification logits are obtained using Equation (Equation 6). By training only a small subset of parameters, APE-T avoids costly fine-tuning of cache model and improves the model performance by updating refined features for both modalities.

### 3.2. Image-Driven Prompt Text Generation

To further optimize the prompt text in APE, i.e., the prior text knowledge of the CLIP, we introduce an image-driven prompt text generation module, as shown in Figure 1 (2). This method’s essence is leveraging Qwen-VL’s fine-grained recognition function to generate image-based prompt texts, improving classification weights compared to text generated solely by language models. To reduce the cost of generating prompt texts, we employ a clustering strategy to select representative images for each category and generate descriptions solely for these representatives. The implementation steps are as follows:Representative image selection: For simplicity, we extract traditional image features for clustering, recognizing the importance of color and texture features in crop disease recognition. These features are highly stable and intuitive, especially in distinguishing subtle and complex disease types. To achieve this, we used the K-means clustering algorithm to select representative images based on both color and texture features. Specifically, we calculate the average color and GLCM (Gray Level Co-occurrence Matrix) texture features for each image, then cluster these combined features using K-means. This approach allows us to cluster each category separately and select *M* representative images for each of the *C* categories to form a collection ${\left\{R_{i}\right\}}_{i=1}^{C}$, where $R_{i}$ denotes the set of representative images belonging to class *i*.Prompt text generation: We sequentially input the selected representative images from each category into Qwen-VL, employing the following unified template command for querying: “**Can you help me describe this [CLASS] leaf?**”, where **[CLASS]** is replaced with the specific disease category. This operation aims at generating detailed descriptions that are closely aligned with the image content. As shown in Figure 2, we show three different categories of crop diseases, the prompt text generated by Qwen-VL with image-driven guidance, and the prompt text generated by the language model GPT3.5 [40] and Qwen-VL without image-driven guidance. For instance, in the image-driven prompt text generated for the “grape leaf blight” category, “dark brown spots” are identified as crucial indicators of the disease, and the generated text also describes additional information about the blade surface. In contrast, without image-driven guidance, although the text generated by GPT3.5 and Qwen-VL also includes relevant disease features, it is not as detailed as the prompt text generated with image-driven input.Integrate text information: We consolidate and organize the collection of textual descriptions $T_{i}\;\left(i=1,2,\ldots ,C\right)$ corresponding to representative images of all categories. The generated prompt texts will serve as an important basis for subsequent classification weights.

### 3.3. Text Feature Fusion in Training-Free (VLCD) Mode

Continuing the training-free strategy of APE, our training-free model VLCD also introduces the PR module and trilateral relationship exploration. Furthermore, we introduce the cross-attention mechanism to process the prompt texts generated in Section 3.2, emphasizing salient features among intra-class prompt texts, as shown in Figure 1 (3). Details are provided as follows:

Suppose there are *C* categories, with each category having prompt *M* texts. For the category *i*, we encode the text descriptions set $T_{i}$ of category *i* obtained in Section 3.2 using CLIP’s text encoder to obtain the embedding vectors $X_{i}\in R^{M\times D}$, where *D* is embedding dimension. Let us denote $S_{i}$ as the attention scores between all prompt texts of category *i*, which is obtained by
(9)$$S_{i}=X_{i}\cdot X_{i}^{T}.$$

Then, we utilize Softmax() to compute the attention weights *A*: (10)$$A=softmax(S_{i}).$$

Next, for the category *i*, we compute its weighted text embedding vectors: (11)$$A_{W}=A*X_{i}.$$

The weighted text embedding vectors $A_{W}$ of the obtained category *i* is averaged to obtain the final text feature vector: (12)$$a_{i}=mean\left(A_{W}\right)\in R^{D}.$$

By stacking the text feature vectors $a_{i}\;\left(i=1,2,\cdots ,C\right)$ of each category, we obtain $W_{C}=\left[a_{1},a_{2},\cdots ,a_{C}\right]\in R^{C\times D}$. Then, we use PR module for significant channel selection to obtain $W_{C}^{\prime }\in R^{C\times E}$.

During testing, we adopt a similar strategy to APE, as shown in part (a) of Figure 1(1), with slight modifications. Specifically, we utilize the $W_{C}$ and $W_{C}^{\prime }$ from the above process to explore the trilateral relation exploration.

First, we use CLIP’s image encoder to obtain $f\in R^{D}$, and then we use cosine similarity representation to represent the relationship between the obtained test image features and the prompt text, for which we substitute W in Equation (Equation 1) with $W_{C}$: (13)$$R_{fW}=fW_{C}^{⊺}\in R^{1\times C}.$$

Next, the PR module is used to obtain $f^{\prime }\in R^{E}$. The relationship between the test image $f^{\prime }$ and training images $F^{\prime }$ is expressed using Equation (Equation 4). In the relationship between training images and prompt texts, we substitute $W^{\prime }$ in Equation (Equation 5) with $W_{C}^{\prime }$: (14)$$R_{F^{\prime }W^{\prime }}=\exp\left(\gamma D_{KL}\left(F^{\prime }{W_{C}^{\prime }}^{T}|L\right)\right)\in R^{1\times CK}.$$

By calculating $R_{F^{\prime }W^{\prime }}$ to quantify the relative importance of each training image feature for prediction accuracy, we can update the cached model knowledge. Finally, we use Equation (Equation 6) to obtain the final classification logits.

### 3.4. Text Feature Enhancement in Training-Required (VLCD-T) Mode

During training, we also follow the strategy of APE-T. To further enhance the key feature weights of inter-class prompt texts, we additionally introduce SE Attention to weight $W_{C}$ obtained in Section 3.3 during training to obtain $W_{\mathrm{att}}$, allowing the model to focus more on critical inter-class prompt text features, as shown in Figure 1 (4). Similarly, we freeze the cache model and make $R_{F^{\prime }W^{\prime }}$ learnable during training. We apply Res to both textual features $W_{\mathrm{att}}$ and training-set features $F^{\prime }$.

Specifically, we process $W_{C}$ using the SE Attention module.

First, for the weight vector $W_{C}$, we perform a global average pooling (*AvgPool*) to obtain a scalar *y*, representing the average value of all features: (15)$$y=AvgPool(W_{C}).$$

Then, *y* is transformed and activated to obtain the attention weights λ: (16)$$\lambda =\sigma (W_{2}\cdot ReLU\left(W_{1}\cdot y\right)),$$
where $W_{1}$ and $W_{2}$ are two learnable linear transformation matrices, σ and ReLU are activation functions. The first linear layer $W_{1}$ reduces the feature dimension from *D* to D/r, where *r* is the reduction ratio. The ratio controls the dimensionality of the intermediate representation, facilitating efficient dimensionality reduction while preserving essential information from the original input. The second linear layer $W_{2}$ restores it to the original dimension. The activation functions ensure that the output values are confined between 0 and 1, representing attention weights for each channel.

Finally, the original weight vector $W_{C}$ is weighted using the weights λ to obtain $W_{\mathrm{att}}$: (17)$$W_{\mathrm{att}}=\lambda \cdot W_{C}.$$

Through the SE Attention processing, we have weighted the original weight vector, increasing the emphasis on discriminative inter-class prompt text vector features.

During testing, we conducted the same trilateral relationship exploration as APE-T, as shown in Figure 1 (1) part (b). First, we use CLIP’s image encoder to encode the test image to obtain $f\in R^{D}$, and then obtain $f^{\prime }\in R^{E}$ through the PR module. Next, we use $W_{\mathrm{att}}$ as W in Equation (Equation 7). The score between $f^{\prime }$ and $F^{\prime }$ is computed using Equation (Equation 8), and, finally, the model’s ultimate logits are obtained using Equation (Equation 6).

## 4. Experiment

### 4.1. Settings

#### 4.1.1. Dataset

The dataset used in our study integrates the Plant Village dataset [41] with self-expanded data. Plant Village, meticulously constructed by Hughes et al., is a large-scale image repository widely used in the global plant pathology community, encompassing over 50,000 high-quality images showcasing diverse states of 14 crop species under the influence of 17 fungal diseases, 4 bacterial diseases, 2 viral diseases, 2 oomycete diseases, and 1 moth infestation. Plant Village mainly uses pictures collected in a laboratory environment; in order to increase the richness of the images, we intentionally augment the dataset by incorporating publicly available images covering 16 novel diseases and additional categories, sourced from a wide variety of backgrounds, including multiple crop samples and diverse laboratory and field settings. The resulting consolidated dataset consists of 54 disease classes and comprises a total of 57,495 images, as detailed in Table 1. Figure 3 presents sample data, with the top row displaying representative images from Plant Village and the second row featuring newly added non-laboratory-environment samples.

#### 4.1.2. Implementation Details

To validate the effectiveness of our method on few-shot classification tasks, we conduct comprehensive experiments. First, to ensure the model’s generalization ability and prevent overfitting, we randomly divided the images of each category in the collected dataset into training, validation, and test sets in an 8:1:1 ratio. The training set is used to support few-shot learning tasks, the validation set is used to search for the optimal parameter combination, and the test set is used for model performance evaluation. In the few-shot classification experiments, we randomly select images from the training set based on the settings of 1, 2, 4, 8, and 16 images per class for training, and perform parameter search and model performance evaluation on the full validation and test sets, respectively. In both the training-free VLCD and training-required VLCD-T setups, we adopt ResNet-50 [42] as the visual encoder and Bert [43] as the text encoder to build the CLIP backbone. We obtain the pre-trained weights of both encoders from CLIP [3] and freeze them during training. Data preprocessing adhered to CLIP standards [3], encompassing random cropping, scaling, and horizontal flipping. The training-required setting of VLCD-T includes batch size is 256, learning rate is 0.001, and the optimizer is AdamW [44] with a cosine scheduler. The setting of the PR module’s parameters followed the optimal setting in APE [7]. Furthermore, the number of prompt texts per class is set to 10, the training epochs are set to 30, and the reduction parameter in the SE Attention module is set to 32.

### 4.2. Performance Analysis

Since most current few-shot crop disease classification methods focus primarily on single-image modality analysis, we chose to compare the classification accuracy of several mainstream fine-tuned CLIP methods on our dataset to facilitate the analysis of our proposed method’s effectiveness. The experimental results are shown in Table 2.

In the training-free setting, we compare three methods that leverage cache model for inference without requiring additional training: Tip-Adapter [6], APE [7], and our VLCD. Tip-Adapter utilizes the standard CLIP prompt “**A photo of a [CLASS]**”, whereas APE and VLCD employ multi-text prompt ensembles. APE adopts an average ensemble of multi-text, whereas VLCD employs a cross-attention mechanism for weighted multi-text integration. According to the experimental results, VLCD exhibits higher accuracy across all few-shot settings. For example, VLCD’s classification accuracy is 1.25% higher than that of APE under 1-shot setting, indicating the effectiveness of the image-driven prompt text generation module and cross-attention mechanism.

In the training-required setting, we compare VLCD-T with five methods that require training: CoOp [21], KgCoOp [23], CLIP-Adapter [24], Tip-Adapter-F [6], and APE-T [7]. Our VLCD-T outperforms previous methods in all few-shot settings and achieves the highest accuracy. Notably, when the few-shot setting is 1 or 2, VLAD-T’s classification accuracy improves by 4.73% and 3.89%, respectively, compared to using APE-T. These results convincingly demonstrate that VLCD-T can more effectively capture critical and discriminative features from inter-class prompt texts, thereby enhancing the overall performance of the model.

### 4.3. Ablation Study

In this section, we explore the effectiveness of our method through extensive ablation experiments.

#### 4.3.1. Different Prompt Texts

To evaluate the impact of prompt text quality on image classification, we conduct comparative experiments examining the effects of varying prompt generation strategies and the quantity of prompt texts on classification performance. We compare the performance differences between the language model GPT3.5 and the multimodal model Qwen-VL in answering questions with and without image input. Both GPT3.5 and Qwen-VL without image inputs are tasked with generating prompts for different categories using a consistent question format: “**Help me generate K sentences describing this [CLASS]**”. In the image-driven scenario of Qwen-VL, it is provided with an image along with the question to identify and generate relevant prompts.

The experimental results are shown in Figure 4. We compare the influence of different numbers of prompts (2, 4, 6, 8, and 10) under various few-shot settings (0, 1, 2, 4, 8, and 16) when employing three distinct prompt generation strategies on classification effectiveness.

In all few-shot settings, the accuracy of image-driven prompts generated by Qwen-VL for classification outperforms that of non-image-driven prompts generated by GPT3.5 and Qwen-VL, regardless of the chosen prompt number, in both training-free and training-required settings. This outcome underscores the significant positive effect of visual information on generating high-quality prompts that enhance model’s classification ability.

Under different few-shot settings, we observe varying impacts of different numbers of prompt texts on few-shot learning. In lower few-shot settings (e.g., 0, 1, and 2), fewer prompts (e.g., 2) generated by GPT3.5 and Qwen-VL without image-driven guidance led to higher classification accuracy compared to when more prompts are used. This could be because, without images, the limited amount of generated prompts are often more precise and of higher quality; increasing the number of prompts may introduce redundant information, potentially diluting the quality of the prompts and negatively impacting the classification performance. Conversely, when Qwen-VL generates prompts with image-driven guidance for classification, its accuracy increases as the number of prompts grows in cases where the few-shot setting is 0, 1, or 2. This is because the input images provide additional visual information, allowing Qwen-VL to better comprehend and generate descriptive sentences related to the categories, thus improving the quality of the prompts. However, when the few-shot is larger (e.g., 4, 8, or 16), the improvement in classification accuracy due to increasing the number of prompts became less pronounced across all methods. This could be because, in these cases, as the number of training shots increases, the model can capture more visual feature information purely from images, making additional prompt texts less impactful on boosting classification performance.

#### 4.3.2. Representative Image Selection Strategy

To validate the effectiveness of clustering in selecting representative images, we conduct few-shot classification experiments using Qwen-VL for prompt texts generation, with inputs of randomly selected images and images selected based on clustering in both training-free and training modes. Experimental results are shown in Table 3.

In the training-free setting, we compare the cases with shots of 0, 1, 2, 4, 6, 8, and 16. In the training-required setting, we compare the cases with shots of 1, 2, 4, 6, 8, and 16. The results indicate that, under both the training-free and training-required settings, the classification accuracy increases with more shots. When the shot number is small, employing representative images selected via clustering to generate prompt texts exhibits a more pronounced advantage. For example, in the training-free setting, the classification accuracy of prompt texts generated by randomly selecting pictures is 21.62% under 0-shot setting, whereas utilizing cluster-selected images leads to an improved accuracy rate of 22.59%. As the number of shot increases, the difference in accuracy rates between the use of cluster-selected images for generating prompt texts and randomly chosen images diminishes; however, the former consistently maintains a marginally higher performance throughout. This observation substantiates that the methodology of inputting cluster-selected images into Qwen-VL to generate prompt texts for classification purposes is indeed more efficacious.

#### 4.3.3. Effectiveness of Attention Mechanisms

To validate the effectiveness of the attention mechanisms in processing prompt texts, we conduct a series of experiments examining model performance in few-shot classification. We compare the difference between direct averaging used in APE [7] and our cross-attention in processing intra-class prompt text vectors, as well as the impact of including or excluding the SE Attention module on inter-class prompt text vectors. Experimental results and settings are detailed in Table 4.

In the training-free setting, the cross-attention method significantly outperforms the averaging method across all few-shot settings, particularly when the number of few-shots is small, indicating that cross-attention is more effective at extracting key features from intra-class prompt texts that are crucial for classification decisions.

During training, we observed that, regardless of whether SE Attention is used to process inter-class prompt texts during subsequent training, the method of using cross-attention to handle intra-class prompt text vectors resulted in improved classification accuracy compared to using the average method. In addition, we found that, regardless of whether average or cross-attention is used for processing intra-class prompt text vectors, the classification performance during training is better when using SE Attention for processing inter-class prompt text vectors compared to methods without SE Attention. It is worth noting that, when cross-attention and SE Attention are used simultaneously, the accuracy increases from 53.43% in the first training setting without any attention mechanism to 58.16% under 1-shot, further confirming the crucial role of cross-attention and SE Attention in adjusting the model’s classification weights.

Additionally, we also investigate the relevant parameters of the SE Attention module during training, focusing on the influence of parameter *r* (reduction) as well as the training epoch on the crop disease classification accuracy.

Parameter *r* adjusts the SE Attention activation function, changing the attention weight distribution and the model’s attention to the prompt texts. Table 5 shows that increasing *r* is positively correlated with improved classification accuracy, confirming that adjusting *r* can enhance attention to key text features and optimize disease image classification. When *r* is set to 32, the model achieves the highest classification accuracy across all few-shot settings. The SE Attention module with a larger *r* can more effectively guide the model to focus on key information, improving classification performance in limited sample scenarios.

Table 6 shows the classification accuracy with and without SE Attention module under different training epochs (from 10 to 50). The results show that, regardless of the epoch setting, the experimental group utilizing SE Attention consistently outperforms the control group that does not employ the module. Notably, the benefits of using SE Attention are more pronounced under low epoch values and a small number of shots. For example, when epoch is 10 and shot number is 1, compared with the 41.44% classification accuracy achieved by the control group that does not use the SE Attention, the experimental group that applies this module achieves a classification accuracy of 50.41%, with an accuracy improvement of 8.97%. These findings robustly demonstrate that our SE Attention module can significantly enhance the model’s classification efficiency.

#### 4.3.4. Different Network Backbones

To comprehensively verify the effectiveness of our method, we compare the performance under different network backbones (ResNet-50 [42], ResNet-101 [42], VIT-B /16 [43], and VIT-B /32 [43]) and performance of various methods in few-shot crop disease classification. We focus on the performance of our methods (VLCD and VLCD-T) with the baseline methods (Tip-Adapter [6], Tip-Adapter-F [6], APE [7], and APE-T [7]) under two settings (training-free and training with 30 epochs) for accuracy changes. The experimental results are shown in Figure 5, where our method outperforms the other methods under all four backbones.

## 5. Visualization

### 5.1. Performance Comparison of Different Prompt Text Generation Strategies

We study the impact of three different prompt text generation strategies on specific crop classification recognition: one is to use the language model GPT3.5 to generate non-image-driven prompt texts, another is to generate non-image-driven prompt texts using Qwen-VL’s text answering, and the last is to utilize Qwen-VL’s image comprehension to generate image-driven prompt texts. Figure 6 shows the top five classification prediction results of each strategy on three specific crop test images, including the top five possible categories predicted along with the probability associated with each prediction. The figure reveals that, among the three specific categories of test images shown, when using the image-driven generated prompt text for classification, its top five prediction categories and accuracy are significantly better than the other two non-image-driven prompts text generation methods. When all three methods correctly predict the category, such as “coffee rust” in the table, the predicted probability using the image-driven prompt text for classification is also higher than the other two methods. This finding underscores that the integration of image information in prompt text creation leads to improved generalization within model testing, thereby offering more reliable and precise predictions in practical applications.

To gain deeper insights into how the model effectively focuses on the relevant visual features during the processing of prompt texts, we borrow the Grad-CAM (Gradient-weighted Class Activation Mapping) [45] visualization technology. Analogous to the work by Shen et al. [46], we adopt a heat map approach to display the image–text relevance, visually revealing the attention distribution that the model allocates to the critical feature areas within the crop disease imagery after receiving the textual prompts. In our experiment, we examine the effects of three different prompt text generation strategies on the model performance. For each test image, corresponding to its given prompt text, we employ Grad-CAM to visualize the attention distribution over the image following the receipt of these text prompts. The experimental results are shown in Figure 7. We find that, compared with the other two non-image-driven prompt text generation strategies, the image-driven prompt text generated by Qwen-VL can more accurately focus on the salient features pertaining to the provided descriptions.

### 5.2. Dynamic Changes in Accuracy under the SE Attention Module

We closely monitor and record the dynamic changes in the accuracy of the training and test sets over time after the introduction of the SE Attention module in the training process, as shown in Figure 8. This experiment focuses on the few-shot crop disease classification where the shot numbers are set to 1, 2, 4, and 8 and the training epoch is set to 30.

The observations from Figure 8 indicate that the training accuracy gradually increases with iterations when not using SE Attention during training, although at a relatively modest pace. However, when SE Attention is introduced, a significant improvement in the training accuracy is observed in the early stages of training, followed by a subsequent slowdown in the rate of increase, yet generally maintaining a level above that observed without SE Attention. In the testing stage, while the test accuracy steadily improves without SE Attention, the extent of the improvement is somewhat limited. Conversely, with the inclusion of SE Attention, the test accuracy exhibits a marked enhancement during the initial stages of training, indicating that the incorporation of the SE Attention module promotes efficient learning in the model and boosts its classification efficiency.

## 6. Summary

In this paper, we propose a few-shot crop disease classification algorithm that integrates VLMs and attention mechanisms to address the limitations of the traditional classification methods in agriculture. By incorporating VLMs, prompt text generation, and attention mechanisms, our method significantly enhances both the precision and efficiency of disease classification. The experimental results demonstrate outstanding performance under few-shot tasks, substantiating its practical value in real-world applications.

Nevertheless, our method still has some shortcomings. During the prompt text generation process, to avoid designing a general domain question template without images, we only used simple questions. Our future work will focus on increasing the diversity of the question templates. Currently, we have only used the free and open-source Qwen-VL and GPT3.5. We plan to try more models in the future, such as GPT4, to further validate our approach. Additionally, we employed a plug-and-play lightweight SE Attention mechanism during training, and we will continue to explore more attention mechanisms in subsequent work. Another limitation is the use of the cache model, which, while reducing the training time, requires providing all known category images, thus limiting its generalization to unknown categories.

In conclusion, this study suggests promising potential for the intelligentization of agricultural production. The algorithm appears to have broad application prospects in the agricultural field and may offer new avenues for innovation in related multimodal domains. In the future, it is expected to be extended to fields such as botany, ecology, and environmental protection, where it can be combined with professional knowledge to develop diverse applications, making significant contributions to achieving sustainable agricultural development and environmental protection goals.

## Figures and Tables

**Figure 1 sensors-24-06109-f001:**
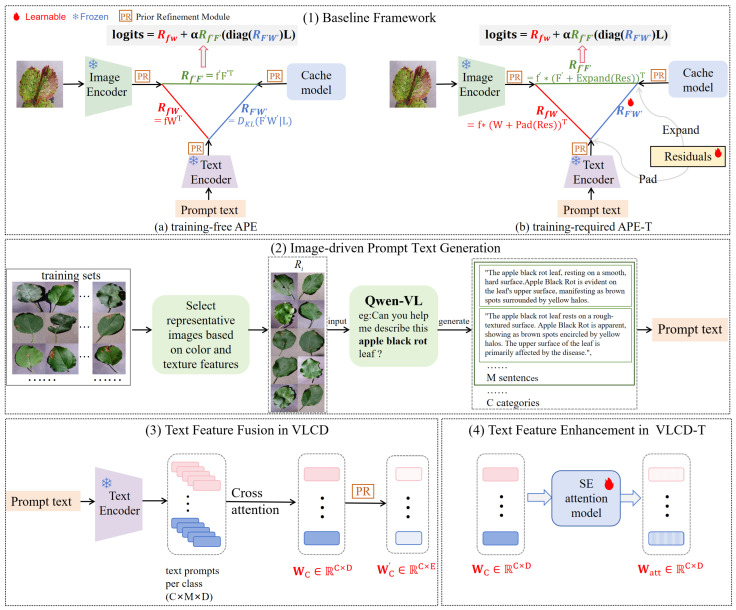
Overall flowchart. (**1**) Shows the baseline framework, where the left side (**a**) is the training-free APE and the right side (**b**) is the training-required APE-T. (**2**) Shows the image-driven prompt text generation module. (**3**) Shows the text feature fusion process in VLCD, where $W_{C}$ corresponds to W of APE in (**1**), and $W_{C}^{\prime }$ corresponds to $W^{\prime }$ of APE and APE-T in (**1**). (**4**) Shows the text feature enhancement process in VLCD-T, where $W_{\mathrm{att}}$ corresponds to W of APE-T in (**1**).

**Figure 2 sensors-24-06109-f002:**
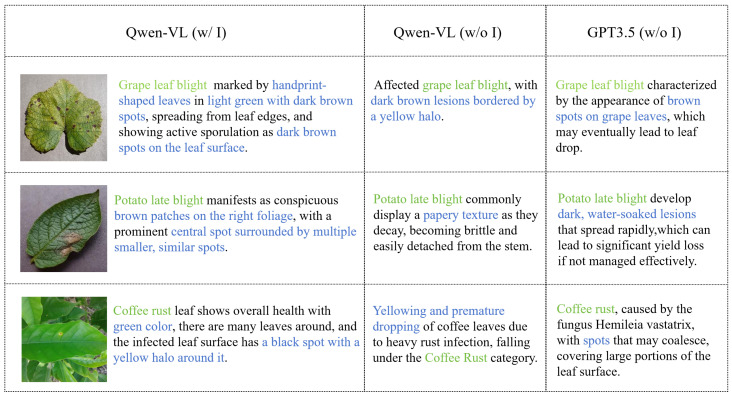
Examples of prompt texts generated by different methods. The green font highlights the class name, and the blue font highlights the distinctive visual feature. The first column includes the prompt texts generated by Qwen-VL with image-driven guidance, the second column includes the prompt texts generated by Qwen-VL without image-driven guidance, and the third column includes the prompt texts generated by GPT3.5 without image-driven guidance.

**Figure 3 sensors-24-06109-f003:**
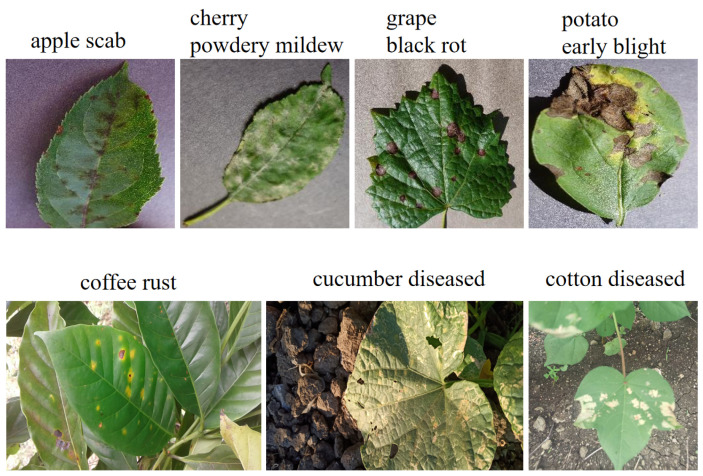
Examples of some images in the crop disease dataset.

**Figure 4 sensors-24-06109-f004:**
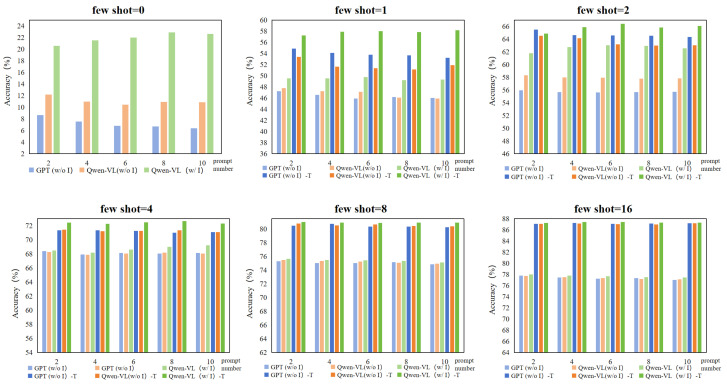
Performance of different prompt texts in few-shot classification tasks. “w/o I” means non-image-driven prompts; “w/ I” means image-driven prompts. The first column of the legend represents the training-free mode, and the second column “-T” represents the training-required mode. The ordinate represents the classification accuracy (%), and the abscissa represents the number of prompt texts.

**Figure 5 sensors-24-06109-f005:**
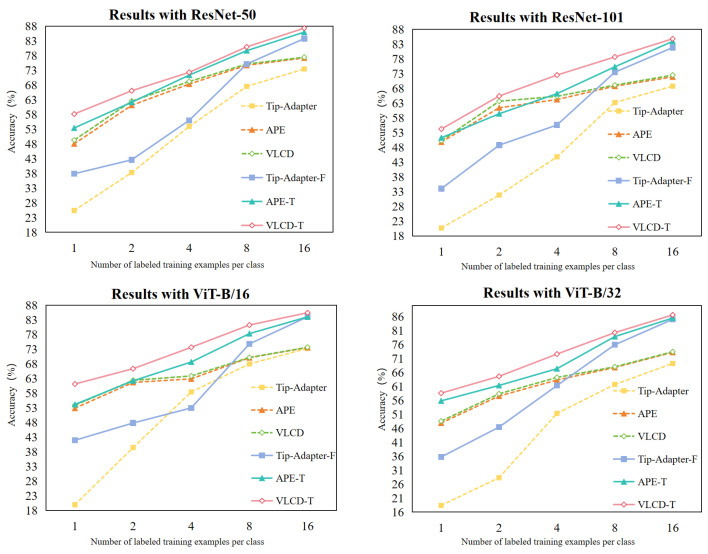
Ablation studies with different backbones. Dashed and solid lines indicate training-free and training-required methods, respectively.

**Figure 6 sensors-24-06109-f006:**
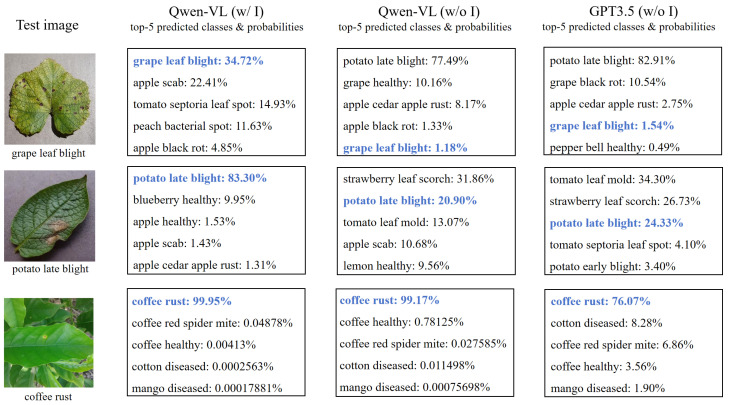
Top 5 classification prediction categories and accuracy (%) of different prompt texts on specific crop test images. “w/o I” means without image-driven guidance; “w/ I” means with image-driven guidance. The blue bold font represents the predicted correct category and probability.

**Figure 7 sensors-24-06109-f007:**
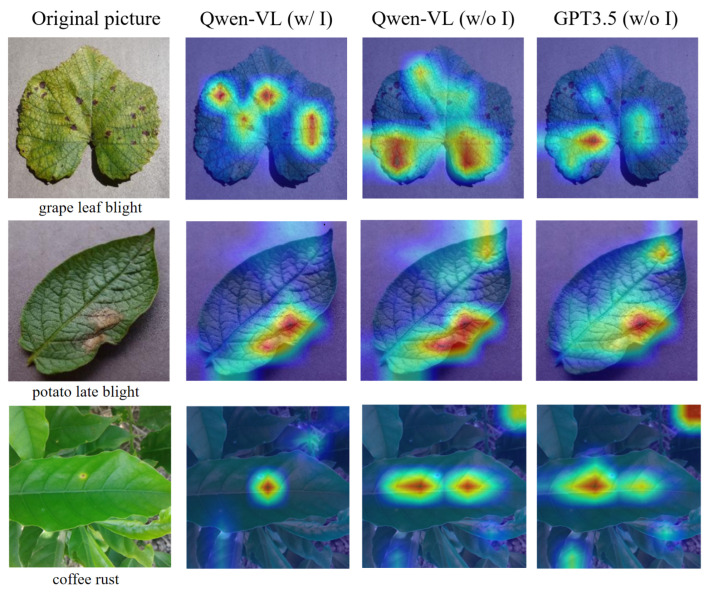
Heat map of correlation between different prompt texts and test images. The first column shows the original image, the second column displays the heat map of the correlation between the image and the image-driven prompt text generated by Qwen-VL, the third column shows the correlation heat map between the image and the non-image-driven prompt text generated by Qwen-VL, and the fourth column shows the correlation heat map between the image and the non-image-driven prompt text generated by GPT3.5.

**Figure 8 sensors-24-06109-f008:**
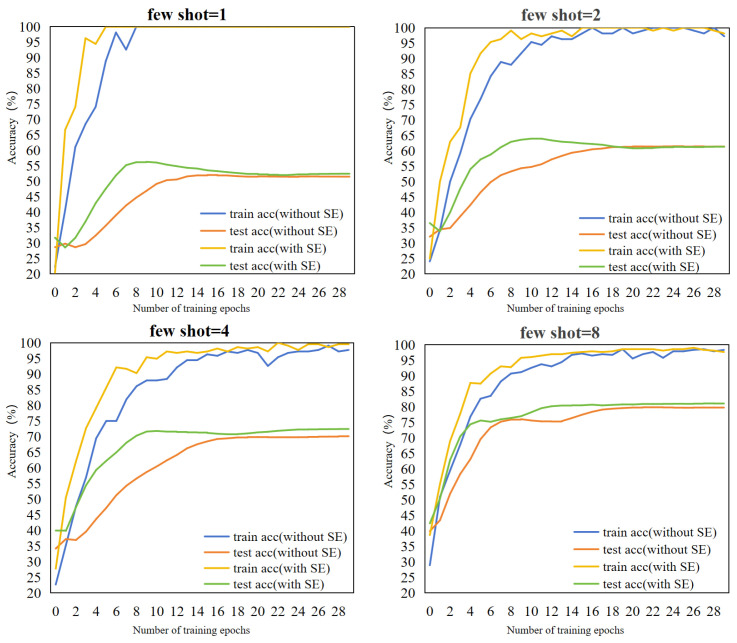
Dynamic evolution of the SE Attention mechanism on training and test set accuracy over training epochs for different few-shot settings. “without SE” represents not using SE Attention; “with SE” represents using SE Attention. The horizontal coordinate is the training epoch and the vertical coordinate is the classification accuracy. “train acc” represents the training accuracy and “test acc” represents the test accuracy.

**Table 1 sensors-24-06109-t001:** Categories and quantity distribution of crop diseases. The non-bolded text indicates existing categories in the Plant Village, while the bolded font signifies the added categories.

Disease Category	Number of Images	Disease Category	Number of Images
apple black rot	621	strawberry leaf scorch	1109
apple cedar apple rust	275	tomato bacterial spot	2127
apple healthy	1645	tomato early blight	1000
apple scab	630	tomato healthy	1591
blueberry healthy	1502	tomato late blight	1915
cherry healthy	854	tomato leaf mold	952
cherry powdery mildew	1052	tomato mosaic virus	373
corn cercospora leaf spot gray leaf spot	513	tomato septoria leaf spot	1771
corn common rust	1192	tomato spider mites	1676
corn healthy	1162	tomato target spot	1404
corn northern blight	985	tomato yellow leaf curl virus	3357
grape black rot	1180	**coffee healthy**	**282**
grape black measles	1383	**coffee red spider mite**	**136**
grape healthy	423	**coffee rust**	**282**
grape leaf blight	1076	**cotton diseased**	**288**
orange huanglongbing	5507	**cotton healthy**	**427**
peach bacterial spot	2297	**cucumber diseased**	**227**
peach healthy	360	**cucumber healthy**	**241**
pepper bell bacterial spot	997	**lemon diseased**	**67**
pepper bell healthy	1491	**lemon healthy**	**149**
potato early blight	1000	**mango diseased**	**255**
potato healthy	152	**mango healthy**	**159**
potato late blight	1000	**pomegranate diseased**	**261**
raspberry healthy	371	**pomegranate healthy**	**277**
soybean healthy	5090	**rice bacterial leaf blight**	**40**
squash powdery mildew	1835	**rice brown spot**	**40**
strawberry healthy	456	**rice leaf smut**	**40**

**Table 2 sensors-24-06109-t002:** Classification accuracy (%) of different fine-tuning strategies on crop disease dataset.

Few-Shot Setup	1	2	4	8	16
Zero-shot CLIP [3]: 13.72					
**Training-free**					
Tip-Adapter [6]	25.42	38.31	53.95	67.54	73.46
APE [7]	48.06	61.13	68.34	74.70	77.20
VLCD	49.31	62.55	69.20	75.14	77.44
**Training-required**					
CoOp [21]	43.44	42.01	64.14	70.79	85.59
KgCoOp [23]	25.49	27.52	32.26	25.49	55.24
CLIP-Adapter [24]	19.38	20.76	20.76	33.53	52.86
Tip-Adapter-F [6]	37.89	42.62	56.02	75.14	83.76
APE-T [7]	53.43	62.18	71.38	79.68	85.99
VLCD-T	58.16	66.07	72.32	80.95	87.31

**Table 3 sensors-24-06109-t003:** Accuracy (%) of prompt texts generated by different representative image selection strategies when used for few-shot classification.

Few-Shot Setup	0	1	2	4	8	16
**Training-free**						
Random selection	21.62	48.96	61.41	67.98	75.08	77.39
Cluster selection	22.59	49.31	62.55	69.20	75.14	77.44
**Training-required**						
Random selection	-	57.83	65.72	72.19	80.65	87.08
Cluster selection	-	58.16	66.07	72.32	80.95	87.31

**Table 4 sensors-24-06109-t004:** Classification accuracy (%) of different prompt text processing methods on the crop disease dataset. “Without SE” represents not using SE Attention, “With SE” represents using SE Attention, and “✓” represents using this method.

	Intra-Class Prompt Text Processing		Inter-Class Prompt Text Processing		Accuracy in Different Few-Shot Setups					
**Setup**	**Average**	**Cross-Attention**	**Without SE**	**With SE**	**0**	**1**	**2**	**4**	**8**	**16**
Training-free	✓	-	-	-	20.80	48.06	61.13	68.34	74.70	77.20
	-	✓	-	-	22.59	49.31	62.55	69.20	75.14	77.44
Training-required	✓	-	✓	-	-	53.43	62.18	71.38	79.68	85.96
	-	✓	✓	-	-	54.04	62.35	71.47	79.88	86.26
	✓	-	-	✓	-	56.61	65.14	72.06	79.96	86.71
	-	✓	-	✓	-	58.16	66.07	72.32	80.95	87.31

**Table 5 sensors-24-06109-t005:** Classification accuracy (%) on crop disease dataset under different parameters *r* during training.

Few-Shot Setup	1	2	4	8	16
*r* = 2	56.36	64.57	70.02	78.58	84.91
*r* = 4	56.89	65.03	71.08	79.33	85.18
*r* = 8	58.13	66.04	70.68	78.51	84.98
*r* = 16	55.56	65.20	69.97	79.90	84.91
*r* = 32	58.16	66.07	72.32	80.95	87.31

**Table 6 sensors-24-06109-t006:** Classification accuracy (%) on crop disease dataset at different training epochs under training settings. Under different epoch settings, “without SE” represents not using SE Attention, “with SE” represents using SE Attention, and the third row represents the accuracy difference between using and not using SE Attention.

	Few-Shot Setup	1	2	4	8	16
epoch = 10	without SE	41.44	55.40	59.46	70.70	78.30
	with SE	50.41	62.37	67.35	76.06	82.75
		+8.97	+6.97	+7.89	+5.36	+4.45
epoch = 20	without SE	52.49	60.76	68.73	76.06	82.49
	with SE	58.01	66.06	71.87	79.88	86.14
		+5.52	+5.30	+3.14	+3.73	+2.68
epoch = 30	without SE	54.04	62.35	71.07	79.68	86.26
	with SE	58.16	66.07	72.32	80.95	87.31
		+4.12	+3.72	+1.25	+1.27	+1.05
epoch = 40	without SE	54.48	62.35	71.35	80.36	86.76
	with SE	58.36	65.62	74.45	82.10	88.00
		+3.88	+3.13	+3.10	+1.74	+1.24
epoch = 50	without SE	54.44	63.00	72.39	81.28	87.60
	with SE	58.16	65.88	74.89	82.49	88.16
		+3.72	+2.88	+2.50	+1.21	+0.56

## Data Availability

The data utilized in this study are composed of several publicly available datasets, which have been combined to form a new dataset. The combined dataset is available at the following link: https://drive.google.com/drive/data/Plant_leave_diseases_dataset (accessed on date 29 July 2024).

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
