# Peer review of "Few-Shot Image Classification of Crop Diseases Based on Vision–Language Models"

_sensors, 2024, doi:10.3390/s24186109_

Round 1
Reviewer 1 Report
Comments and Suggestions for Authors
Thank you for letting me know about your research.
In “Few-Shot Image Classification of Crop Diseases Based on Vision-Language Models,” the authors demonstrate the effect of integrating image and text modalities in overcoming the limitations of current unimodal approaches in accurately identifying complex crops diseases in the face of data scarcity.
Having read the manuscript, I have no objections regarding its publication conditional upon resolving the following issues.
Issues to be addressed:
- The work is interesting, as it addresses two emerging research fields: multimodal language models (which in this work incorporates both image and text information) and model-centric vs. data-centric AI (a concept introduced by Andrew Ng in 2021). The manuscript would benefit from a statement linking this work to the current debate on model-centric AI and data-centric AI. Although you integrate an existing dataset (Plant Village) with self-expanded one, your work, with leveraging Qwen-VL and fine-tunning CLIP, falls under model-centric AI, which underscores the importance of further developing model-centric approaches. I suggest adding the following text to the contributions’ part within the ‘Introduction’ Section (pages 2 and 3, lines 76 through 91): “With reference to the ongoing debate on model-centric AI and data-centric AI [1], our work belongs to the category of model-centric AI, highlighting the significance of further developing model-centric approaches [2].”
[1] Ng, A (2021). AI Doesn’t Have to Be Too Complicated or Expensive for Your Business. 2021. Available online.
[2] Hamid, O. H. (2023). Data-Centric and Model-Centric AI: Twin Drivers of Compact and Robust Industry 4.0 Solutions. Applied Sciences, 13(5), 2753.
- Editing issues, for example, typos, grammatical mistakes, punctuation, and naming conventions:
o In Section 2 (Related Work), page 3, line 100, the in-text citation of reference 7 should be ‘Irmak and Saygili’ instead of ‘IRMAK et al.’
- Please go through all listed references within the Section ‘References’ and ensure complete information of the cited works. For example, the journal of reference 48 is missing. The same applies to reference 31.
- In Subsection 2.4 (Attention Mechanism), the authors cite a work (reference 31) as reference for attention mechanisms. However, attention mechanisms as method for leveraging deep learning (DL) networks was first introduced by Vaswani et al. (2017), prior to the work cited in reference 31. Therefore, I highly recommend including Vaswani et al.’s work in the list of references.
[3] Vaswani, A., Shazeer, N., Parmar, N., Uszkoreit, J., Jones, L., Gomez, A. N., ... & Polosukhin, I. (2017). Attention is all you need. Advances in neural information processing systems, 30.
- Figure 9 on page 21: Though the caption describes the numbers within the horizontal axes of the plots, it would be better to include a title for the horizontal axes at least in two lower plots, corresponding to few shot = 4 and 8.
Author Response
- Comments and Suggestions:
The work is interesting, as it addresses two emerging research fields: multimodal language models (which in this work incorporates both image and text information) and model-centric vs. data-centric AI (a concept introduced by Andrew Ng in 2021). The manuscript would benefit from a statement linking this work to the current debate on model-centric AI and data-centric AI. Although you integrate an existing dataset (Plant Village) with self-expanded one, your work, with leveraging Qwen-VL and fine-tunning CLIP, falls under model-centric AI, which underscores the importance of further developing model-centric approaches. I suggest adding the following text to the contributions’ part within the ‘Introduction’ Section (pages 2 and 3, lines 76 through 91): “With reference to the ongoing debate on model-centric AI and data-centric AI [1], our work belongs to the category of model-centric AI, highlighting the significance of further developing model-centric approaches [2].”
[1] Ng, A (2021). AI Doesn’t Have to Be Too Complicated or Expensive for Your Business. 2021. Available online.
[2] Hamid, O. H. (2023). Data-Centric and Model-Centric AI: Twin Drivers of Compact and Robust Industry 4.0 Solutions. Applied Sciences, 13(5), 2753.
Answer by Authors:
Thank you for your suggestive comments. We have added the relevant text to the "Introduction" section on page 3 and updated the references on page 24. The modifications in the revised manuscript are highlighted in yellow.
- Comments and Suggestions:
Editing issues, for example, typos, grammatical mistakes, punctuation, and naming conventions:
In Section 2 (Related Work), page 3, line 100, the in-text citation of reference 7 should be ‘Irmak and Saygili’ instead of ‘IRMAK et al.’
Answer by Authors:
Thank you for your careful reading and valuable comments. We have corrected the in-text citation in Section 2 (Related Work) on page 3 to ‘Irmak and Saygili’ . Similar issues are also corrected in the citation of Reference 14 on page 3, with the modifications highlighted in yellow in the revised manuscript.
- Comments and Suggestions:
Please go through all listed references within the Section ‘References’ and ensure complete information of the cited works. For example, the journal of reference 48 is missing. The same applies to reference 31.
Answer by Authors:
Thank you for your careful readings and valuable suggestions. Reference 31 had a citation error previously. We have carefully reviewed all listed references and updated References [31] and [48] to include complete information. Similar errors have also been corrected, with the modifications highlighted in yellow in the reference section of the revised manuscript.
- Comments and Suggestions:
In Subsection 2.4 (Attention Mechanism), the authors cite a work (reference 31) as reference for attention mechanisms. However, attention mechanisms as method for leveraging deep learning (DL) networks was first introduced by Vaswani et al. (2017), prior to the work cited in reference 31. Therefore, I highly recommend including Vaswani et al.’s work in the list of references.
Answer by Authors:
Thank you for your valuable comments. We originally cited the work of Vaswani et al. (2017) in Reference 39, however, due to a previous citation error, we have now moved the work of Vaswani et al. (2017) to Reference 31 and made corresponding corrections to the original citation of Reference 39 in the revised manuscript. The changes are highlighted in yellow in Section 2.4 on page 5 of the revised manuscript. The new Reference 39 has been corrected to a new supplementary reference.
- Comments and Suggestions:
Figure 9 on page 21: Though the caption describes the numbers within the horizontal axes of the plots, it would be better to include a title for the horizontal axes at least in two lower plots, corresponding to few shot = 4 and 8.
Answer by Authors:
Thank you for your suggestion regarding Figure 9. We have added the title “Number of training epochs” to the horizontal axis in the relevant plots. Additionally, based on Reviewer 3's comments and subsequent modifications, Figure 9 has been renumbered to Figure 8. We have ensured all references to this figure are updated accordingly in the manuscript.
Thank you very much!
Reviewer 2 Report
Comments and Suggestions for Authors
Summary:
This work focuses on crop disease classification and employs the multimodal model Qwen-VL to generate detailed textual descriptions for disease images, improving the quality of classifier weights. By integrating cross-attention and SE attention mechanisms, this approach significantly improves classification accuracy in few-shot scenarios, offering a valuable tool for agricultural pathology and smart farming.
Strength:
1. The paper is well-organized and easy to follow.
2. The experiments are extensive, and the results are promising.
Weakness:
1. No need to show Figure 1.
2. Could the authors explain the motivation to use Qwen-VL? Is it the optimal model to generate text for leaf datasets? Is it possible to also try other alternatives, e.g., what’s the performance of GPT-4 in Figure 3?
3. The motivate to use cross-attention and SE attention should be given and emphasized.
4. There are some related works for few-shot learning with VLMs that should be discussed, e.g., “Task Residual for Tuning Vision-Language Models, CVPR2023”, “GraphAdapter: Tuning Vision-Language Models with Dual Knowledge Graph, NeurIPS2023” and “Beyond sole strength: Customized ensembles for generalized vision-language models, ICML2024”.
5. The information of reference should be up to date, e.g., “Tip-adapter: Training-free clip-adapter for better vision-language modeling” is published on ECCV 2022. Please update the similar information.
Comments on the Quality of English LanguageThe paper is well-organized and easy to follow.
Author Response
- Comments and Suggestions:
No need to show Figure 1.
Answer by Authors:
Thank you very much for your valuable suggestion. We have removed Figure 1 from the manuscript, and the numberings of the remaining figures throughout the updated manuscript have been modified accordingly.
- Comments and Suggestions:
Could the authors explain the motivation to use Qwen-VL? Is it the optimal model to generate text for leaf datasets? Is it possible to also try other alternatives, e.g., what’s the performance of GPT-4 in Figure 3?
Answer by Authors:
Thank you for pointing this out.
We used Qwen-VL considering the following two reasons: 1) its effectiveness in handling fine-grained images like sick leaf images, which is crucial for our leaf disease classification task; 2) it is free and open-source, allowing us to conduct lots of experiments in a cheaper way. To validate the optimality, we should try all of the existing vision-language large models, including the non-open-source ones, which could be impracticable. Besides, we should note that we are aiming to demonstrate that it is important to introduce image information while generating text prompt. As demonstrated by the public benchmark results (https://github.com/QwenLM/Qwen-VL), Qwen-VL is already one of the most effective vision-language large models. As such, the experimental results (Page 14-15, Section 4.3.1) by using Qwen-VL could be convincing.
Nonetheless, following your suggestion, we will try more models in the future, such as GPT-4 (if we can get the authorization to use it), to further enrich our experimental results.
We have added relevant discussions in Section 2.3 (Automatic Generation of Prompt Text) on page 5 and Section 6 (Summary) on page 21. All of these modifications are highlighted in yellow in the revised manuscript.
- Comments and Suggestions:
The motivate to use cross-attention and SE attention should be given and emphasized.
Answer by Authors:
Thank you very much for your valuable comments. Our motivation for using the cross-attention and the SE attention mechanism is to optimize the feature weight distribution when the model processes the prompt text, thereby adjusting the classification weight of the model and enhancing the classification accuracy. In addition, the cross-attention and SE attention mechanism can be easily integrated into the model without significantly increasing development costs or training time. This point has been emphasized in Section 2.4 (Attention Mechanism), page 5 of the revised manuscript. All of the above modifications are highlighted in yellow in the revised manuscript.
- Comments and Suggestions:
There are some related works for few-shot learning with VLMs that should be discussed, e.g., “Task Residual for Tuning Vision-Language Models, CVPR2023”, “GraphAdapter: Tuning Vision-Language Models with Dual Knowledge Graph, NeurIPS2023” and “Beyond sole strength: Customized ensembles for generalized vision-language models, ICML2024”.
Answer by Authors:
Thank you very much for your valuable comments. We have added the discussion of these related works in Section 2.2 (Vision-Language Models and Fine-tuning) on page 4 and updated the references on page 24., with the additions highlighted in yellow in the revised manuscript.
- Comments and Suggestions:
The information of reference should be up to date, e.g., “Tip-adapter: Training-free clip-adapter for better vision-language modeling” is published on ECCV 2022. Please update the similar information.
Answer by Authors:
Thank you very much for your valuable comments. We have checked the references and updated the reference information to ensure it up to date. We have also checked and corrected other similar errors, highlighting the references in yellow in the revised manuscript for correction.
Thank you very much!
Reviewer 3 Report
Comments and Suggestions for Authors
The authors used the multimodal model Qwen-VL to generate meticulous textual descriptions for representative disease images, which can capture fine-grained and image-specific information better, thereby enhancing prompt quality. The cross-attention and the SE (Squeeze-and-Excitation) attention are integrated into the training-free mode VLCD (Vision-Language model for Crop Diseases classification) and the training-required mode VLCD-T (VLCD-Training) respectively for prompt text processing, which can improve the performance of disease classification. The experimental results showed the effectiveness of the method. The manuscript has a certain degree of novelty in the crop diseases classification
But there some problems should be solved.
1) Representative image selection is based on clustering. What the cluster algorithm is used?
2) The authors used the relative Large Language model, such as Qwen-VL,to generate text prompts and to use them in crop diseases classification. Is fine tuning needed for this Large Language model?
3) In the dataset description, training and testing descriptions, I cannot find how to realize fine-tuning CLIP. And how to select training samples and testing samples for few shot classification? I also cannot find that.
Author Response
- Comments and Suggestions:
Representative image selection is based on clustering. What the cluster algorithm is used?
Answer by Authors:
Thank you for your valuable comments. We used the K-Means clustering algorithm to select representative images based on both color and texture features. Specifically, we calculate the average color and GLCM (Gray Level Co-occurrence Matrix) texture features for each image, then cluster these combined features using K-Means. The relevant explanation have been added to Section 3.2 (Image-driven Prompt Text Generation), page 9 of the revised manuscript, with the modifications highlighted in yellow.
- Comments and Suggestions:
The authors used the relative Large Language model, such as Qwen-VL,to generate text prompts and to use them in crop diseases classification. Is fine tuning needed for this Large Language model?
Answer by Authors:
Thank you for your valuable comments. In our work, we did not perform fine-tuning on the Qwen-VL model for our specific use. Qwen-VL can directly generate detailed text descriptions from images and simple questions using the pre-trained model, streamlining the process and reducing the need for additional training. This explanation has been added to Section 2.3 (Automatic Generation of Prompt Text), page 5 of the revised manuscript, with the modifications highlighted in yellow.
- Comments and Suggestions:
In the dataset description, training and testing descriptions, I cannot find how to realize fine-tuning CLIP. And how to select training samples and testing samples for few shot classification? I also cannot find that.
Answer by Authors:
Thank you for your valuable comments.
We used a similar method to APE to fine-tune the CLIP model and constructed a cache model through a small sample training set. We introduced VLCD (Vision-Language model for Crop Diseases classification) that does not require additional training and VLCD-T (VLCD-Training) that requires training. In the training-free VLCD mode, we use image descriptions generated by Qwen-VL as prompt texts and extract text features using CLIP's text encoder. These features are then processed using a cross-attention mechanism to generate classification weights. In the training-required VLCD-T mode, we further introduce the SE attention mechanism to handle prompt text vectors for different categories during training, emphasizing key features. VLCD-T allows fine-tuning of class residuals for visual features and textual representations, as well as adjustments to cache scores. This information can be found in Section 3 (Methods) on page 5-12 of the revised manuscript.
We randomly divided the images of each category in the collected dataset into training, validation, and test sets in an 8:1:1 ratio. The training set is used to support few-shot learning tasks, the validation set is used to search for the optimal parameter combination, and the test set is used for model performance evaluation. In the few-shot classification experiments, we randomly select images from the training set based on the settings of 1, 2, 4, 8, and 16 images per class for training, and perform parameter search and model performance evaluation on the full validation and test sets, respectively. In the description of the dataset, we only introduced the dataset's source, the disease categories it includes, the number of images, and the dataset construction process, but in the training details of Section 4.1.2 (Implementation Details) on page 13, we explained in detail how to divide the training set, validation set, and test set, as well as their functions.
Thank you very much!